# The active-zone protein Munc13 controls the use-dependence of presynaptic voltage-gated calcium channels

Nathaniel Calloway, Géraldine Gouzer, Mingyu Xue, Timothy A Ryan*

Department of Biochemistry, Weill Cornell Medical College, New York, United States

**Abstract** Presynaptic calcium channel function is critical for converting electrical information into chemical communication but the molecules in the active zone that sculpt this function are poorly understood. We show that Munc13, an active-zone protein essential for exocytosis, also controls presynaptic voltage-gated calcium channel (VGCC) function dictating their behavior during various forms of activity. We demonstrate that in vitro Munc13 interacts with voltage-VGCCs via a pair of basic residues in Munc13's C2B domain. We show that elimination of this interaction by either removal of Munc13 or replacement of Munc13 with a Munc13 C2B mutant alters synaptic VGCC's response to and recovery from high-frequency action potential bursts and alters calcium influx from single action potential stimuli. These studies illustrate a novel form of synaptic modulation and show that Munc13 is poised to profoundly impact information transfer at nerve terminals by controlling both vesicle priming and the trigger for exocytosis.

## Introduction

The active-zone protein, Munc13, plays a central and essential role in all known forms of chemical synaptic transmission (*Augustin et al., 1999*; *Varoqueaux et al., 2002*). Munc13 is critical for correct assembly of exocytic proteins in preparation for neurotransmitter release that it executes at least in part through interactions with plasma membrane SNARE protein, syntaxin (*Ma et al., 2012*). This large multi-domain protein additionally binds several other key active-zone proteins, including RIM, ELKS and bassoon as well as calmodulin. Munc13, additionally, contains three C2 domains that can mediate interactions with lipid membranes. The importance of Munc13 in synapse function was established in genetic ablation experiments in mice, flies, and worms, but the roles of numerous putative interactions of Munc13 with potential binding partners remain an area of intense interest. Synaptic transmission relies on two distinct molecular pathways for neurotransmitter release: the preparation of neurotransmitter-filled synaptic vesicles to a docked state at the active-zone and action potential-driven opening of voltage-gated calcium channels (VGCCs) causing rapid elevation of intracellular calcium in the vicinity of these vesicles. Although a number of active-zone proteins (Rim, Bassoon, Elks) have been found to play a role in controlling the location and/or abundance of VGCCs (*Kittel et al., 2006*; *Han et al., 2011*; *Davydova et al., 2014*), these potential interactions are not thought to impact VGCC properties themselves. Additionally, different variants of Munc13 have been shown to differentially impact exocytosis depending on the distance between release sites and VGCCs (*Hu et al., 2013*; *Zhou et al., 2013*). Here, we provide compelling evidence that Munc13 interacts with VGCCs in a way that controls calcium channel use-dependence on millisecond to second time scales at nerve terminals. We pinpointed a critical interaction site to 2 basic residues within the C2B domain of Munc13 on a face that is orthogonal to the potential membrane-interacting loops of this domain. Loss of Munc13 at hippocampal nerve terminals profoundly alters the response of VGCCs during brief AP bursts of very high-frequency firing. Although re-expression of Munc13-harboring point mutations that

*For correspondence: taryan@med.cornell.edu

**Competing interests:** The authors declare that no competing interests exist.

**eLife digest** The electrical signals that neurons use to rapidly transmit information along their length cannot cross the gaps—called synapses—that separate one neuron from the next. Instead, the signals trigger the release of chemicals called neurotransmitters, which then stimulate a corresponding electrical signal in a neighboring neuron.

Inside the neuron, the neurotransmitters are packaged into structures called vesicles and are released across the synapse when the vesicle merges with the cell membrane at a location called the active zone. Calcium ions move through proteins known as calcium channels, which are embedded in the neuron's cell membrane in the active zone, and cause the vesicle to merge with the neuron's membrane and release its contents into the synapse.

A protein called Munc13 is also important for helping to release neurotransmitters, which it does by binding to various other proteins in the active zone known to be critical for the process of allowing the vesicle and cell membranes to merge. Now, Calloway et al. have found that Munc13 also interacts with the calcium channels. The experiments used genetic tools to eliminate or mutate Munc13 in rat neurons. Electrical impulses were then applied to these neurons and the flow of calcium ions was monitored at the synapses.

The results showed that Munc13 controls when the calcium channels open and close in response to nerve impulses. Further experiments revealed the specific region of the Munc13 protein that interacts with the calcium channels. Mutations to this part of Munc13 affected the ability of the calcium channels to open and close.

The results indicate that active zone proteins such as Munc13 can potentially play multiple roles in controlling neurotransmitter release. It seems unlikely that Munc13 is the only calcium channel partner that helps sculpt information transfer at synapses. Future studies could investigate how multiple partners work together to determine the behavior of calcium channels in specific locations and at specific times, and how this interplay affects how well synapses work in the brain.

prevent interaction with VGCCs in Munc13-KD synapses restores exocytosis, it does not rescue the alterations in VGCC function. As a result, synapses expressing this mutant Munc13 have profound changes in ultra-fast plasticity of the exocytic response as well. Thus, in addition to its central importance in controlling SNARE assembly Munc13 also tunes temporal aspects of VGCCs and in turn influences ultra-fast plasticity at nerve terminals.

## Results

### Munc13 interacts with VGCCs in vitro

Munc13 isoforms contain numerous protein–protein and protein–ligand interaction domains. These include a $Ca^{2+}$ and lipid-interacting C2B domain homologous to the C2B domains of synaptotagmin-1 and RIM1, which are known to interact with the synaptic protein interaction (synprint) region on $Ca_V2.2$ (*Sheng et al., 1997*; *Chapman et al., 1998*; *Coppola et al., 2001*) (*Figure 1A*). Since Munc13 is known to be targeted to regions of the presynaptic membrane rich in VGCCs (*Weimer et al., 2006*), we wondered if Munc13 might also interact with VGCCs via this conserved C2B domain. We carried out in vitro co-precipitation assays using glutathione S-transferase (GST) Munc13-1 C2B domain fusions and epitope-tagged derivative of the synprint region of $Ca_V2.2$. We showed that the C2B domain of Munc13, like that in synaptotagmin and RIM, also interacts with the synprint region of $Ca_V2.2$ in a $Ca^{2+}$-independent manner (*Figure 1B*). Munc13 isoforms carry a short polybasic sequence in their C2B domain similar to the synprint interaction motif on synaptotagmin that is orthogonal to the lipid-interaction face of the protein (*Figure 1C*). Mutating this polybasic sequence (Munc13-KR/AA) disrupted the in vitro interaction between Munc13-1 and the synprint region of $Ca_V2.2$ (*Figure 1B*) similar to what has been reported for both the synaptotagmin C2B-synprint and RIM C2B-synprint interactions. These findings indicate that the polybasic region in the C2B domains of these proteins is a shared motif that potentially mediates interaction with VGCCs in situ. Previous reports of C2B-synprint interactions have not included functional assays of the effect of the

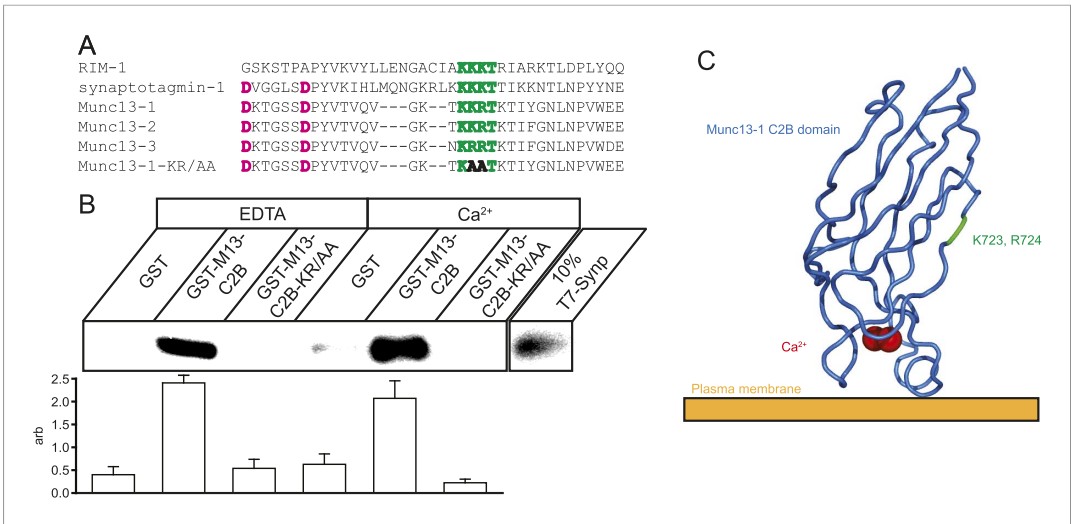

**Figure 1**. Munc13 C2B domain interacts with Ca$_V$2.2 synprint. (**A**) Alignment of C2 domains from RIM1, synaptotagmin-1, Munc13 isoforms 1, 2, and 3, and Munc13-KR/AA showing the synprint binding polybasic sequence (green), Ca$^{2+}$-interacting residues (pink), and the two mutated residues in Munc13-KR/AA to prevent synprint binding (bold). (**B**) Top, representative pull-down of T7-tagged Ca$_V$2.2 Synprint region by Glutathione–sepharose bound GST or GST-Munc13-1-C2B fusions in the presence of 1 mM EDTA or 1 mM Ca$^{2+}$. Bottom, quantification of N = 4 independent pull-down experiments normalized to average blot intensity. Results are mean ± SEM. p < 10$^{-6}$ for grouping by genotype, p = n.s. for the presence of Ca$^{2+}$ or genotype–Ca$^{2+}$ interaction. (**C**) Cartoon illustrating the relative position and orientation of the synprint binding site and the residues mutated in Munc13-KR/AA with respect to the plasma membrane and Ca$^{2+}$ binding site. PDB structure 3KWU.

interaction on presynaptic Ca$^{2+}$ influx. We therefore took advantage of the ability to monitor Ca$^{2+}$ influx at active zones under conditions that perturb the potential Munc13-VGCC interaction.

## Loss of Munc13 decreases action potential-driven presynaptic calcium influx

Although a number of studies have successfully used heterologous expression systems to examine how small modulatory proteins, such as calmodulin and βγ subunits of G proteins, control voltage-gated calcium currents at the plasma membrane this approach is more difficult with large proteins whose subcellular targeting is restricted to specialized compartments, as is the case for Munc13 targeted to active zones. To test for the possible significance of the Munc13–VGCC interaction, we therefore examined the impact of removing Munc13 on action potential-driven calcium influx in nerve terminals. We recently showed that expression of an shRNA targeting the major cortical isoforms Munc13-1 and Munc13-2 (Munc13-KD) led to a 94% or greater depletion of this protein at nerve terminals that results in complete block of exocytosis (*Rangaraju et al., 2014*). As expected single AP responses probed using vG-pHluorin are eliminated in Munc13-KD neurons (*Figure 2A*). We previously developed a robust approach for examining AP-driven presynaptic Ca$^{2+}$ influx using AM-ester loaded Fluo5F at presynaptic boutons (*Hoppa et al., 2012*) identified using a fluorescent-protein-tagged presynaptic protein (VAMP2-mCherry). We found that Munc13-KD resulted in a 34% reduction in Ca$^{2+}$ influx following a single action potential as compared to wild-type (WT) cells (*Figure 2C,D*). This effect was fully reversed by rescuing the knockdown with an shRNA-insensitive variant of Munc13-1 (see below). There are several mechanisms by which Munc13 could modify Ca$^{2+}$ influx: loss of Munc13 could change the number of VGCCs on the presynaptic membrane; it could alter the presynaptic waveform, which controls the fraction of VGCCs that open and the temporal envelope of the driving force for Ca$^{2+}$ entry, or it could alter the gating properties of VGCCs themselves. To investigate possible changes to the AP waveform, we used the genetically encoded voltage-sensitive probe archaerhodopsin fused to green fluorescent protein (Arch-GFP) (*Kralj et al., 2012*) expressed either in WT or Munc13-KD neurons to visualize possible changes in the action potential at presynaptic terminals (*Hoppa et al., 2014*). We found that Munc13-KD did not significantly impact the height or full-width at half-maximum (FWHM) of the AP compared to WT

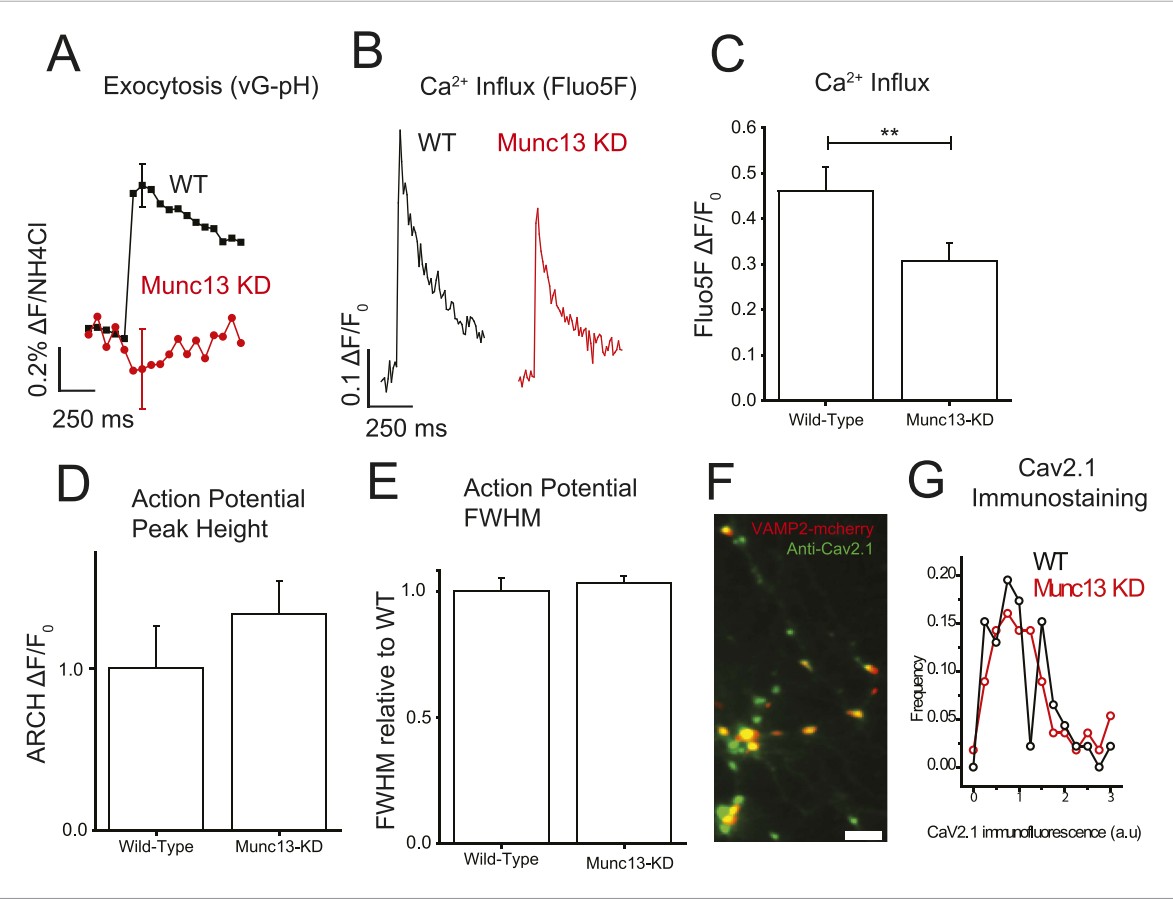

**Figure 2**. Munc13-KD reduces Ca²⁺ influx at the presynaptic terminal. (**A**) Average single AP vG-pHluorin response in wild-type (WT) (black, N = 5) and Munc13-KD (red, N = 5) neurons. (**B**) Example traces of Ca²⁺ influx detected by Fluo5F fluorescence in response to a single AP in WT (black) and Munc13-KD (red) neurons. (**C**) Averages of peak Fluo5F Ca²⁺ measurements ±SEM. N = 10, 13 respectively. **p < 0.01. (**D**) Averages of peak ARCH voltage measurements normalized to WT. N = 5, 7 respectively. Results are mean ± SEM. p = n.s. (**E**) Averages of FWHM from ARCH AP measurements for WT and Munc13 KD normalized to WT. Results are mean ± SEM. N = 5, 7 respectively. p = n.s. (**F**) Example image of VAMP2-mCherry + Munc13-KD synaptic terminals (red) labeled with Alexa488 anti-Ca_v2.1 (green). Scale bar 5 μM. (**G**) Frequency histogram of Cav2.1 staining intensities at VAMP-mCherry-positive Munc13-KD boutons (red) compared to Cav2.1 puncta in non-transfected neurons in the same field of view (N = 56 and 46 boutons for KD and WT respectively from three separate coverslips) showed that ablation of Munc13 does not alter the abundance of Cav2.1 (p = 0.39, 2 tailed t-test).

(*Figure 2D,E*). Based on these data, we concluded that the reduction in Ca²⁺ influx was not the result of changes to the AP waveform. Quantitative immunostaining using an anti-Ca_V2.1 antibody directed to an intracellular domain of this protein (*Hoppa et al., 2012*) showed that the total synaptic abundance of this Ca²⁺ channel isoform was not altered by Munc13-KD (*Figure 2F,G*). At present, however, we lack tools to probe the abundance of the other major VGCC present in these nerve terminals (Ca_V2.2). Immunostaining also does not unambiguously determine the number of VGCCs on the presynaptic plasma membrane vs an intracellular pool since the only available antibodies target intracellular domains of the channels. Thus, the decrease in single AP presynaptic Ca²⁺-influx in the absence of Munc13 could arise from changes in surface abundance of VGCCs or changes in gating properties. To look for possible changes in gating properties, we examined how loss of Munc13 impacted functional plasticity of AP-driven presynaptic Ca²⁺ influx on time scales that would be too fast (ms to s) to involve membrane trafficking events to and from the cell surface.

## Munc13-KD leads to prolonged inactivation of VGCCs after trains of action potentials

To investigate an underlying mechanism of Munc13's control of Ca²⁺ influx in presynaptic boutons that would be separate from changes in VGCC surface abundance, we first examined how the absence of

Munc13 might influence the ability of VGCCs to recover from bursts of activity. Changes in $Ca^{2+}$ influx after the burst would be consistent with alteration in VGCC kinetics of inactivation and/or recovery from protracted stimuli. To examine this issue, we compared single AP $Ca^{2+}$ influx before and after a 0.5-s period of 100 Hz firing (*Figure 3A*). In WT synapses this level of activity did not impact single AP responses measured 0.5 s after the burst compared to before the burst. In contrast in the absence of Munc13, the single AP response 0.5 s after the burst was significantly lower (∼30%) than that measured before the burst (*Figure 3A,B*, *Table 1*). There are two possible explanations for this reduction: (1) in the absence of Munc13 there is increased VGCC inactivation following the 100 Hz stimulus and (2) in the absence of Munc13 VGCCs inactivate to the same extent as in WT but recover more slowly. At present, we do not have the means to separate these possibilities experimentally.

## Munc13-KD reduces a VGCC refractory phase during high-frequency pairs of action potentials

As a second paradigm to study changes in VGCC-gating properties, we probed $Ca^{2+}$ dynamics for paired stimuli on very fast time scales, at the upper limit of stimulation frequencies that could conceivably generate multiple APs (*Staff et al., 2000*). Since presynaptic $Ca^{2+}$ clearance subsequent to AP stimulation occurs over a >100 ms time scale, and as single AP signals are well below the saturation threshold for Fluo5F, the $Ca^{2+}$ influx for an inter-stimulus interval (ISI) of 2 ms should reflect the sum of the $Ca^{2+}$ influx from each stimulus. These experiments showed that in WT synapses the summed response for double AP stimulation was only slightly higher than that obtained with a single AP (*Figure 4A*) indicating a 90% reduction in $Ca^{2+}$ influx on the second AP with a 2 ms ISI (*Figure 4A,G*, *Table 1*). These results are consistent with previous reports that demonstrated profound depression of excitatory postsynaptic current (EPSC) in response to stimulus pairs at these frequencies (*Stevens and Wang, 1995*; *Dobrunz et al., 1997*; *Brody and Yue, 2000*). Those studies independently proposed various hypotheses for the mechanism of this depression including exocytosis-driven inhibition, failure of AP propagation, and inactivation of VGCCs. All three of these possible mechanisms would be potentially consistent with our observations of depression of $Ca^{2+}$ influx for short ISIs. We reasoned that if exocytosis at the active zone leads to a brief refractory period in VGCC function then conditions that

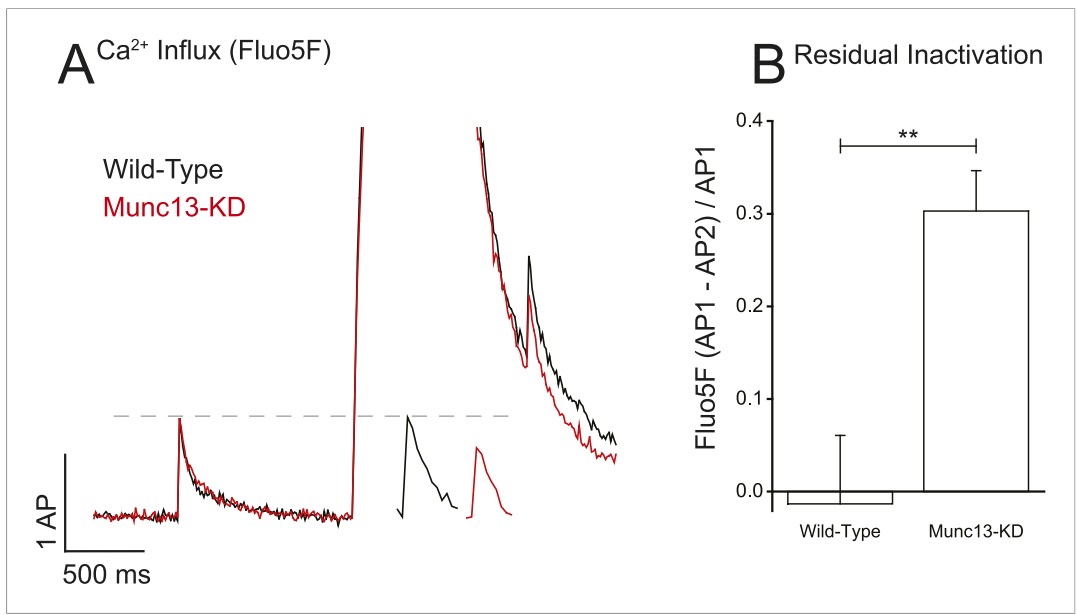

**Figure 3**. Munc13-KD prolongs VGCC inactivation on long-time scales. (**A**) Example of $Ca^{2+}$ reactivation assay for WT and Munc13-KD neurons showing the difference in single AP responses before and after a 100 Hz 50 AP train showing that WT but not Munc13-KD cells have fully recovered from inactivation 500 ms after the end of the train. Inset shows AP following train on expanded time scale for more clarity. (**B**) Average inactivation of $Ca^{2+}$ influx for WT and Munc13-KD neurons from the reactivation assay shown in **A**. Results are mean ± SEM. **p < 0.01.

**Table 1**. Values for Fluo5- and vG-pHluorin-paired pulse experiments

| | | WT | SEM | N | Munc13 KD | SEM | N | Munc13 KD + Munc13-1 | SEM | N | Munc13 KD + Munc13 KR/AA | SEM | N |
|---|---|---|---|---|---|---|---|---|---|---|---|---|---|
| 1 AP Fluo5F | $\Delta F/F_0$ | 0.461 | 0.039 | 10 | 0.306 | 0.039 | 13 | 0.454 | 0.032 | 6 | 0.338 | 0.027 | 11 |
| 2 AP FLuo5F (2 ms ISI) | $\Delta F/F_0$ | 0.511 | 0.052 | 10 | 0.518 | 0.070 | 10 | 0.536 | 0.046 | 6 | 0.518 | 0.044 | 10 |
| AP Post 50 AP Train | $\Delta F/F_0$ | 0.467 | 0.052 | 7 | 0.213 | 0.030 | 10 | 0.444 | 0.039 | 6 | 0.224 | 0.032 | 7 |
| 1 AP vGlut-pHluorin | $\Delta F/NH_4Cl$ | 0.007 | 0.001 | 12 | −0.001 | 0.001 | 5 | 0.004 | 0.001 | 6 | 0.002 | 0.0005 | 7 |
| 2 AP vGLut-pHluorin (2 ms ISI) | $\Delta F/NH_4Cl$ | 0.008 | 0.001 | 12 | – | – | – | 0.004 | 0.001 | 8 | 0.003 | 0.0006 | 8 |

eliminate exocytosis should remove the depression in $Ca^{2+}$ influx. We tested this idea in a manner that is independent of Munc13 by expressing the proteolytic light-chain of tetanus toxin (TeNT-LC) in hippocampal neurons and carried out measurements of AP-driven influx for single and AP pairs at 2 ms ISI. We previously showed that expression of TeNT-LC in these cells eliminates exocytosis (*Rangaraju et al., 2014*). Simply eliminating exocytosis, however, did not change the reactivation behavior of presynaptic VGCCs (*Figure 4B,G*) probed with 2 AP stimulation with a 2 ms ISI, which were indistinguishable from that observed in WT synapses. Thus, the observed fast paired-pulse depression in presynaptic $Ca^{2+}$ influx is not exocytosis driven. In order to determine if the loss of $Ca^{2+}$ influx could result from AP failure, we used Arch-GFP to examine the presynaptic AP waveform for single and AP pairs at 2 ms ISI (*Figure 4C*) measured at individual boutons. Our measurements have all been carried out at 37°C while most of the earlier work examined synaptic depression at colder temperatures (~22–24°C). Arch-GFP-based recordings of AP pairs showed that even at 30°C there is virtually complete failure in AP propagation for the second AP when delivered 2 ms after the first, consistent with the predictions of *Brody and Yue (2000)* (*Figure 4C*). The ratio of the ARCH signal in response to the second stimulus normalized to that of the response to the first stimulus is close to zero, once corrected for the residual signal resulting from the first AP (*Figure 4D*). In contrast at 37°C, similar measurements revealed that the amplitude of the second AP is similar to the first at these short ISIs in both WT and Munc13-KD (*Figure 4C,D*) in all cells examined (*Figure 4D*) (wild-type $AP_2/AP_1 = 1.04 \pm 0.05$ sem, n = 7, Munc13-KD $AP_2/AP_1 = 1.04 \pm 0.16$ sem, n = 7). The width of the second AP was also similar for both wild type (FWHM $AP_2 = 1.2 \pm 0.24$ ms sem, n = 7) and Munc13-KD (FWHM $AP_2 = 1.36 \pm 0.50$ ms sem, n = 7). Therefore, under these conditions both APs successfully propagate to presynaptic boutons and should be equally capable of fully driving VGCCs to the open state in both WT and Munc13-KD boutons. Thus, at physiological temperature, AP failure is not the mechanism of fast synaptic depression of $Ca^{2+}$ influx (or by extension the mechanisms of the depression of exocytosis).

Remarkably, we found that VGCCs behave drastically differently for these fast-time scale paired-pulse stimuli in the absence of Munc13. Munc13-KD nerve terminals displayed much greater $Ca^{2+}$ influx for the second AP relative to the first compared to WT. Following Munc13-KD, the second AP response was 68% of the single AP response (*Figure 4D*, *Table 1*) in stark contrast to cells expressing Munc13 where the second AP response was a mere 10% of the single AP response (*Figure 4A,E*, *Table 1*). These data imply that Munc13 strongly impacts the millisecond time scale dynamics of VGCCs and, in particular, alters the likelihood of either persisting in or entering a refractory state following an AP. Based on this logic, we predicted that due to the low concentration of Munc13 outside of presynaptic boutons, $Ca^{2+}$ influx in other regions of the cell would resemble that in Munc13-KD boutons. Indeed, $Ca^{2+}$ influx recorded at the soma in response to 2 ms spaced pairs of stimuli in WT cells had almost identical properties to presynaptic responses in Munc13-KD cells for *Figure 4F,G*. Our data are thus consistent with the idea that Munc13 actively modulates VGCC properties in the active zone.

We further examined the persistence of this refractory state of VGCCs by examining paired-pulse $Ca^{2+}$ influx over a series of ISIs in both WT and Munc13-KD neurons (*Figure 4H*). WT synapses recovered from reactivation failure by ~ 60% within 5 ms but reached full recovery only for ISIs >40 ms. In contrast, in synapses that lack Munc13, reactivation was much more efficient over the entire 2–40 ms regime (>60%) and was similar to WT values for times >40 ms.

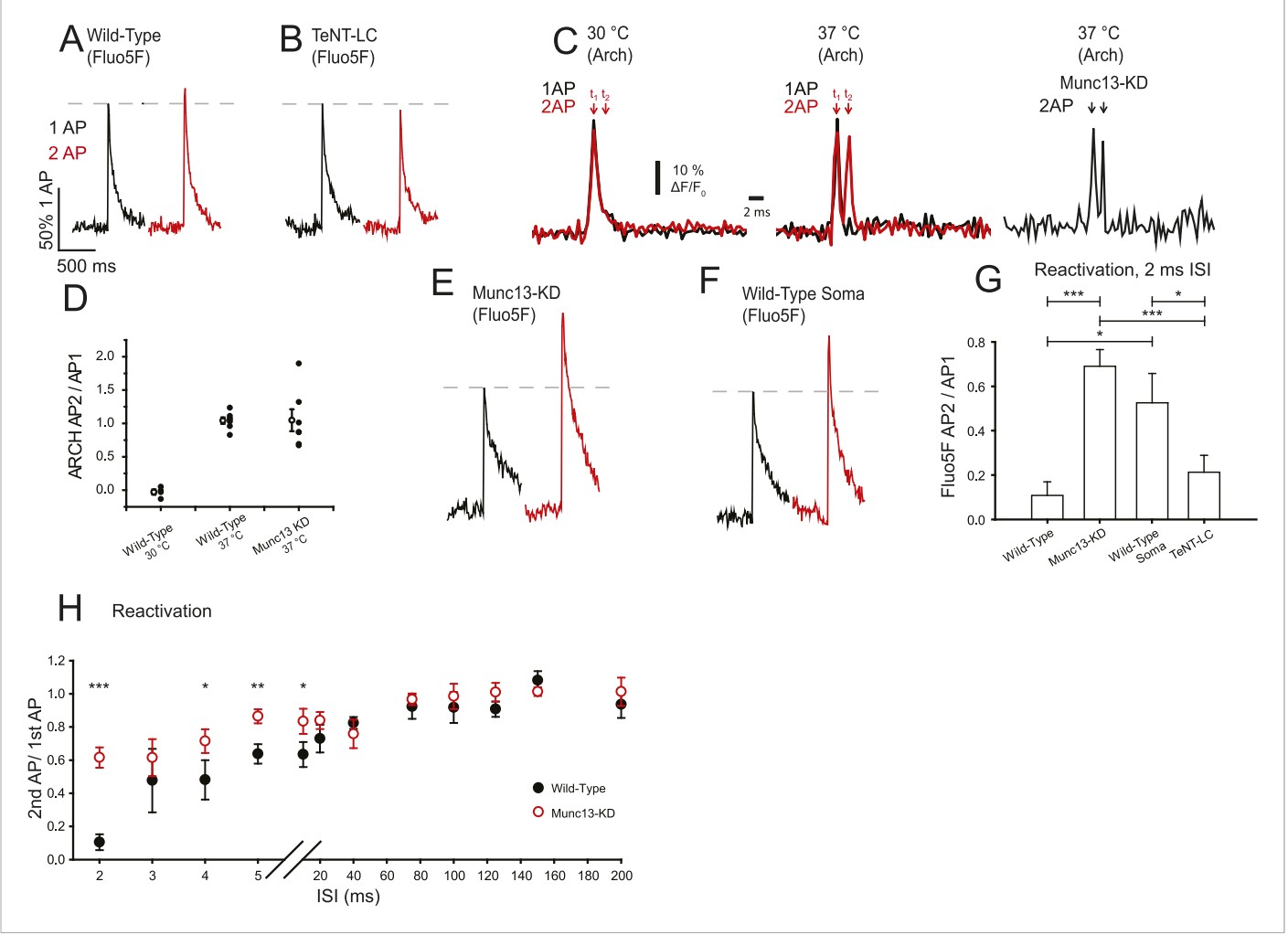

**Figure 4**. Munc13-KD increases VGCC reactivation on a very short time scale. (**A**, **B**, **E**, **F**) Example traces of $Ca^{2+}$ responses to one AP (black) and two APs separated by 2 ms for wild-type (WT) boutons (**A**), TeNT-LC expressing boutons (**B**), Munc13-KD boutons (**E**), and WT soma (**F**). (**C**) Average membrane AP waveform measured using ARCH at WT boutons for single and double AP (2 ms ISI) at 30°C (left) and 37°C (right) indicating AP failure for the second AP at the lower temperature but successful firing and propagation of both APs at physiological temperature. Arch recordings from Munc13-KD boutons also show successful firing of the second AP. (**D**) ARCH amplitude ratios of second AP peak with respect to first AP peak for WT at 30°C (left), WT at 37°C (middle) and Munc13-KD at 37°C (right). Open symbols show the mean ± SEM value for N = 4, WT 30°C; N = 7, WT 37°C, N = 7 Munc13 KD 37°C. (**G**) Average $Ca^{2+}$ signal peak height in 2 ms paired pulses ISI normalized to the respective $Ca^{2+}$ signal peak in response to 1AP. (**H**) Average $Ca^{2+}$ signal peak calculated for the second AP during paired pulses of various ISI for WT (black) and Munc13-KD (red) cells normalized to $Ca^{2+}$ signal peak in response to 1 AP. Results are mean ± SEM. *$p < 0.05$, **$p < 0.01$, ***$p < 0.001$, all other comparisons n.s.

Because knockdown of Munc13-1/2 results in a reduction in single AP $Ca^{2+}$ influx in addition to a reduction in failure to reactivate, a plausible explanation is that fast reactivation is a $Ca^{2+}$-dependent process modulated by $Ca^{2+}$ influx from the first AP. For example, VGCCs are known to undergo $Ca^{2+}$-dependent inactivation (*Lee et al., 2003*; *Chaudhuri et al., 2007*). To investigate if the increased response of the second AP during 2 ms ISI stimulation in Munc13-KD synapses was related to the reduced influx on the first AP, we examined how the failure of VGCCs to reactivate for ISI of 2 ms depended on the magnitude of $Ca^{2+}$ influx during the first AP by varying external $Ca^{2+}$. We measured $Ca^{2+}$ influx in WT synapses for single and pairs of AP stimuli at 2 ms ISI using lower external $Ca^{2+}$ (1.2 mM and 0.6 mM, *Figure 5*), which reduced $Ca^{2+}$ influx to levels seen in Munc13-KD neurons (*Figure 5B*, 39% reduction of single AP for 0.6 mM $Ca^{2+}$), and compared the results to those obtained in 2 mM external $Ca^{2+}$. Our data showed that the failure of VGCC reactivation for short ISI was insensitive to external $Ca^{2+}$ manipulations (*Figure 5A*) and not dependent on the magnitude

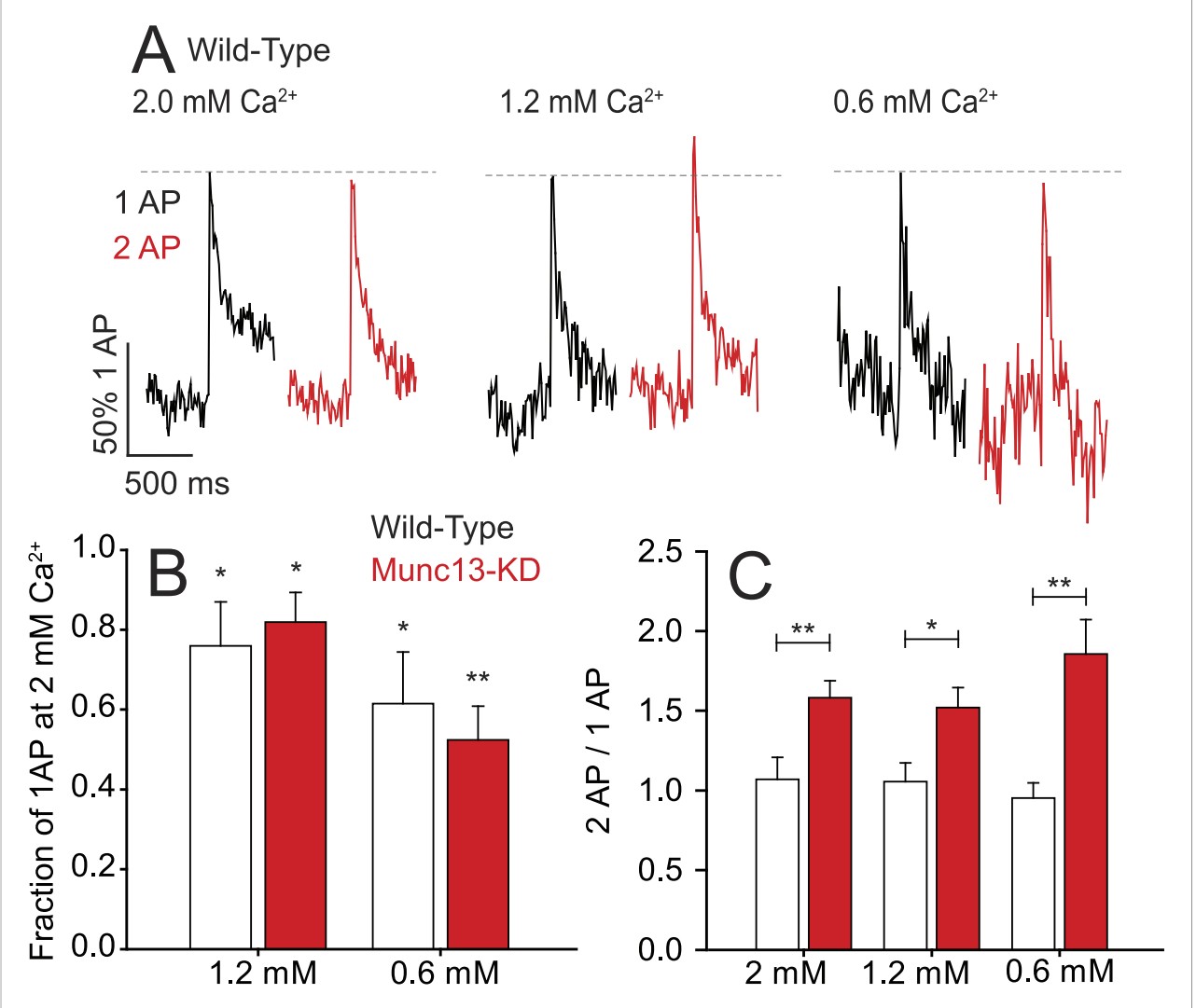

**Figure 5**. Millisecond refractory period controlled by Munc13 is insensitive to extracellular $Ca^{2+}$. (**A**) Example traces of wild-type (WT) neurons stimulated with 1 AP and 2 AP with a 2 ms ISI at 2.0, 1.2, and 0.6 mM extracellular $Ca^{2+}$. Values are normalized to 1 AP for the three extracellular $Ca^{2+}$ conditions. (**B**) Relative $Ca^{2+}$ influx in response to 1 AP upon reduced extracellular $Ca^{2+}$ as a fraction of influx in 2 mM $Ca^{2+}$. N = 5 each. *$p < 0.05$, **$p < 0.01$. Results are mean ± SEM. (**C**) 2 ms ISI $Ca^{2+}$ influx normalized to 1 AP for the three extracellular $Ca^{2+}$ conditions. No significant difference between conditions of the same genotype. N = 5 each. *$p < 0.05$, **$p < 0.01$. Results are mean ± SEM.

of the influx of the first AP (*Figure 5B*). Similarly, the absence of depression in $Ca^{2+}$ influx for the second AP stimulus observed in Munc13-KD synapses was independent of external $Ca^{2+}$ concentration and the amount of influx during the first AP (*Figure 5B,C*). Thus, the refractory period we observe for reactivation of VGCCs following an initial AP is not impacted by changing the magnitude of $Ca^{2+}$ influx in both WT and Munc13-KD neurons. The combination of a slower recovery from inactivation in Munc13-KD neurons, a reduced single AP $Ca^{2+}$ influx and increased reactivation on very fast time scales is consistent with a model whereby the interaction of Munc13 with VGCCs changes the energy barriers between different VGCC states, altering the rates of transitions between these states.

## Mutation of the Munc13 synprint binding motif prevents its effects on presynaptic VGCCs

To test the significance of the observed in vitro interaction between Munc13 and the synprint region of $Ca_V2.2$, we expressed shRNA-resistant derivatives of both Munc13-1 and the mutant lacking

synprint binding capacity, Munc13-KR/AA, in a Munc13-KD nerve terminals. Re-expression of Munc13-1 in Munc13-KD neurons resulted in complete restoration of $Ca^{2+}$ influx from single APs (*Figure 6A,D*). This observation suggests that the effects seen with Munc13-KD are specific to the depletion of Munc13 expression levels as opposed to off target effects. Conversely, in Munc13-KD cells expressing Munc13-KR/AA, $Ca^{2+}$ influx from single APs remained similar to Munc13-KD (*Figure 6A,D*) although exocytosis was largely restored (see below). This result suggests that, indeed, the synprint binding motif in the C2B domain of Munc13 is specifically responsible for the functional effects of Munc13-KD on VGCCs. Similarly, expression of WT Munc13-1 in the KD restored VGCC refractory period observed during 2 ms ISI paired pulses and eliminated the residual inactivation following 50 AP trains whereas expression of Munc13-KR/AA did not (*Figure 6B,C,E,F*). Thus, for these functional assays, rescue with Munc13-1 restored $Ca^{2+}$ dynamics to that of WT cells, whereas Munc13-KR/AA did not and was indistinguishable from Munc13-KD. These data support the idea that the interaction we identified in vitro constitutes the primary molecular determinant of the effects on VGCCs seen in Munc13-KD neurons.

## Munc13-mediated high-frequency depression of VGCC function impacts synaptic vesicle exocytosis

Because we observed a Munc13-mediated reduction in VGCC reactivation with 2 ms ISI-paired pulses, we tested the physiological consequences of such stimuli at the level of presynaptic exocytosis. Bursts of stimulation have been identified at this frequency throughout the nervous system

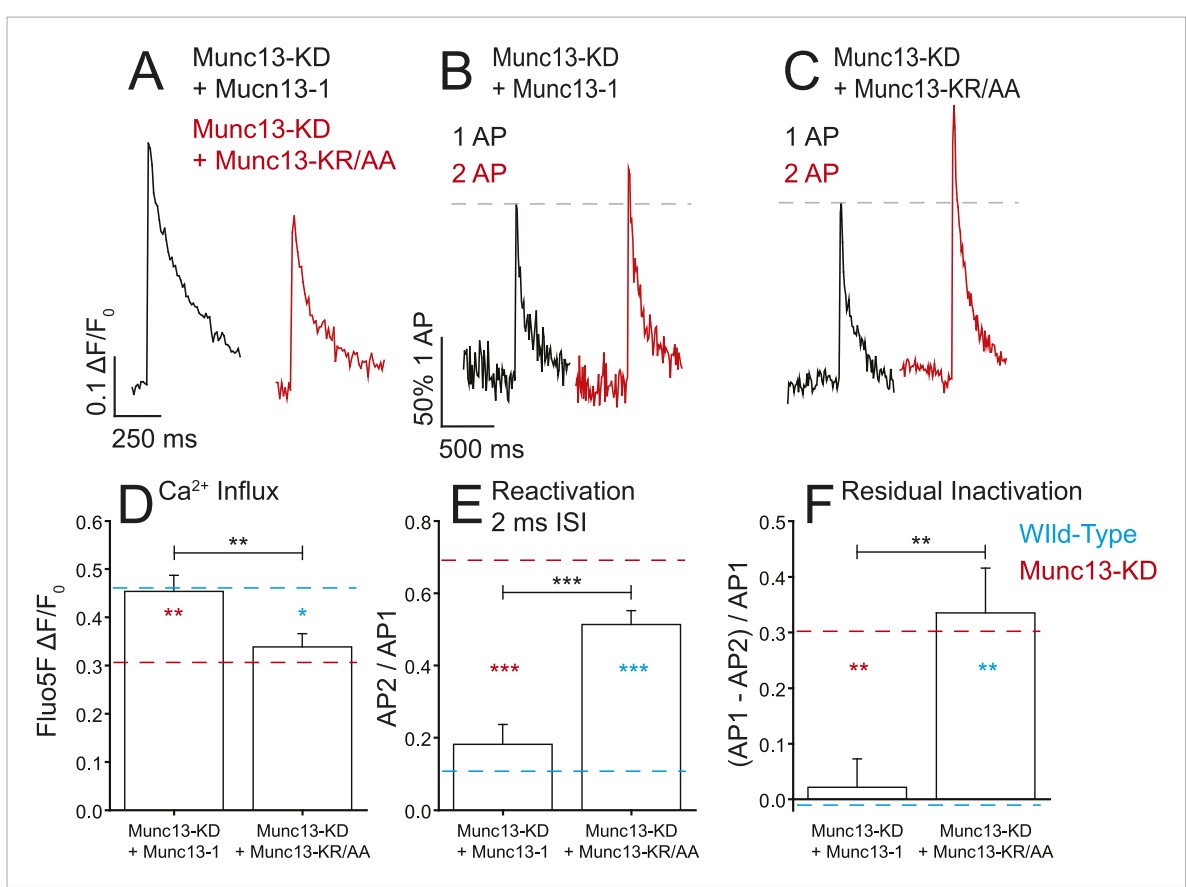

**Figure 6**. Munc13-1 but not Munc13-KR/AA rescues the $Ca^{2+}$ influx phenotypes associated with Munc13-KD. (**A**) Example traces of $Ca^{2+}$ influx detected by Fluo5F fluorescence in response to a single AP in Munc13-1 (black) and Munc13-KR/AA (red) expressing neurons in a Munc13-KD background. (**B**, **C**) Example traces of one AP (black) and two APs separated by 2 ms for Munc13-1 (**B**) and Munc13-KR/AA (**C**) expressing boutons in a Munc13-KD background. (**D–F**) Average values for $Ca^{2+}$ influx (**D**), reactivation at 2 ms ISI (**E**), and residual inactivation (**F**), in Munc13-KD + Munc13-1 and Munc13-KD + Munc 13-KR/AA cells, as in *Figures 3, 4*. Results are mean ± SEM. *p < 0.05, **p < 0.01, ***p < 0.001, all other comparisons n.s.

(*Kandel and Spencer, 1961*; *Llinas and Jahnsen, 1982*; *Gray and McCormick, 1996*), and the fidelity of synaptic transmission in these bursts is strongly influenced by the ISI time scale (*Harris et al., 2001*; *Roberts et al., 2008*). Previous studies have shown that when hippocampal synapses are challenged by pairs of APs separated by only a few milliseconds, the second response is generally profoundly depressed with respect to the first (*Stevens and Wang, 1995*; *Dobrunz et al., 1997*; *Brody and Yue, 2000*). We measured synaptic vesicle exocytosis using vGlut1-pHluorin as previously described (*Voglmaier et al., 2006*; *Ariel and Ryan, 2010*; *Hoppa et al., 2012*; *Ariel et al., 2013*) and compared single AP responses with 2 ms ISI paired pulses in WT cells using the optical reporter. Since endocytosis and re-acidification of vGlut1-pHluorin occurs on a relatively slow time scale (s), the signals from 2 AP with only 2 ms ISI should summate. We found that in WT cells stimulation with two APs at 2 ms ISI resulted in a small (20%) increase in exocytosis over a single AP (*Figure 7A,E*, *Table 1*), corresponding to 80% inhibition of exocytosis from the second AP. This depression in the second AP response was insensitive to decreasing magnitude of influx during the first AP (*Figure 7B,E*), consistent with our observations that reductions in extracellular $Ca^{2+}$ do not affect levels of channel reactivation (*Figure 5*). The complete lack of synaptic vesicle exocytosis precludes measuring fast depression in Munc13-KD cells. However, reintroduction of shRNA-resistant Munc13-1 into the KD background gave very similar fast depression to WT cells (*Figure 7C,E*, *Table 1*). In contrast, expression of Munc13-KR/AA in the Munc13-KD rescued exocytosis from a single AP to levels expected based on $Ca^{2+}$ influx (see below) but failed to restore the profound depression of exocytosis for 2 ms ISI indicating that this mutant effectively decoupled the vesicle priming activity of Munc13-1 from its effects on VGCCs (*Figure 7D,E*, *Table 1*). Synaptic vesicle exocytosis was smaller for Munc13-KR/AA rescue than WT cells as one would predict from the reduced $Ca^{2+}$ influx in Munc13-KR/AA rescues and the steep relationship between $Ca^{2+}$ influx and vesicle exocytosis (*Ariel and Ryan, 2010*). In fact, exocytosis in Munc13-KR/AA rescue in response to single AP was similar to that measured in WT cells in the presence of 1.2 mM extracellular $Ca^{2+}$ (not shown). The fact that VGCC failed to reactivate in 1.2 mM extracellular $Ca^{2+}$ WT cells but not in Munc13-KR/AA rescue suggests that the increase in exocytosis during 2 ms ISI paired pulses in Munc13-KR/AA is a direct effect of the lack of Munc13-VGCC interaction and not a secondary effect of reduced exocytosis. We therefore conclude that this interaction site in Munc13's C2B domain mediates VGCC channel reactivation in nerve terminals.

## Discussion

The observation that Munc13 enhances $Ca^{2+}$ influx and modulates kinetic properties of VGCCs across a range of stimulation conditions has broad implications for understanding how synaptic transmission is controlled. Our data strongly imply that this modulation is mediated by a direct VGCC-Munc13 interaction although we cannot exclude that synprint interaction site on Munc13 C2B influences interactions with other proteins in situ that in turn modulate VGCCs. Nonetheless, we demonstrate that the presence of this critical active-zone protein has potent influence on VGCC function and delineates a new class of mechanisms for sculpting synaptic efficacy. The strong dependence of exocytosis on $Ca^{2+}$ influx and the control of VGCC function on fast time scales imply that synaptic throughput can be actively controlled by such VGCC-active-zone protein interactions. The fact that many presynaptic proteins interact with VGCCs (*Hans et al., 1999*; *Zhong et al., 1999*; *Han et al., 2011*; *Kaeser et al., 2011*; *Liu et al., 2011*) suggests that other cases of this type of VGCC control may be operational at nerve terminals.

Our experiments also provide a mechanistic explanation for the profound synaptic depression observed for ultra-short ISIs. The analysis of Brody and Yue strongly implied that for hippocampal neurons action potentials fail to invade nerve terminals when delivered in rapid succession. Using newly developed fast genetically encoded voltage indicators, we confirmed that for a 2 ms ISI this is the case (*Figure 4C*) but only for the cooler temperatures at which those early experiments were performed. Remarkably, at physiological temperatures synaptic depression for 2 ms ISIs persists (*Figure 7A*), but the root cause is not AP failure (*Figure 4C*) but a Munc13-mediated failure of VGCC reactivation (*Figure 4A*).

Given that Munc13 itself is a multi-domain protein, the rules of governing its interaction with VGCCs might be tunable at different synapses depending on the repertoire of other molecular partners and the Munc13 variant. It is interesting to note that the original description of VGCC's interaction with synaptic proteins (synprint) was that between syntaxin and $Ca_V2s$ (*Rettig et al., 1997*; *Sheng et al., 1997*). Syntaxin is also a primary interacting partner of Munc13 suggesting that that

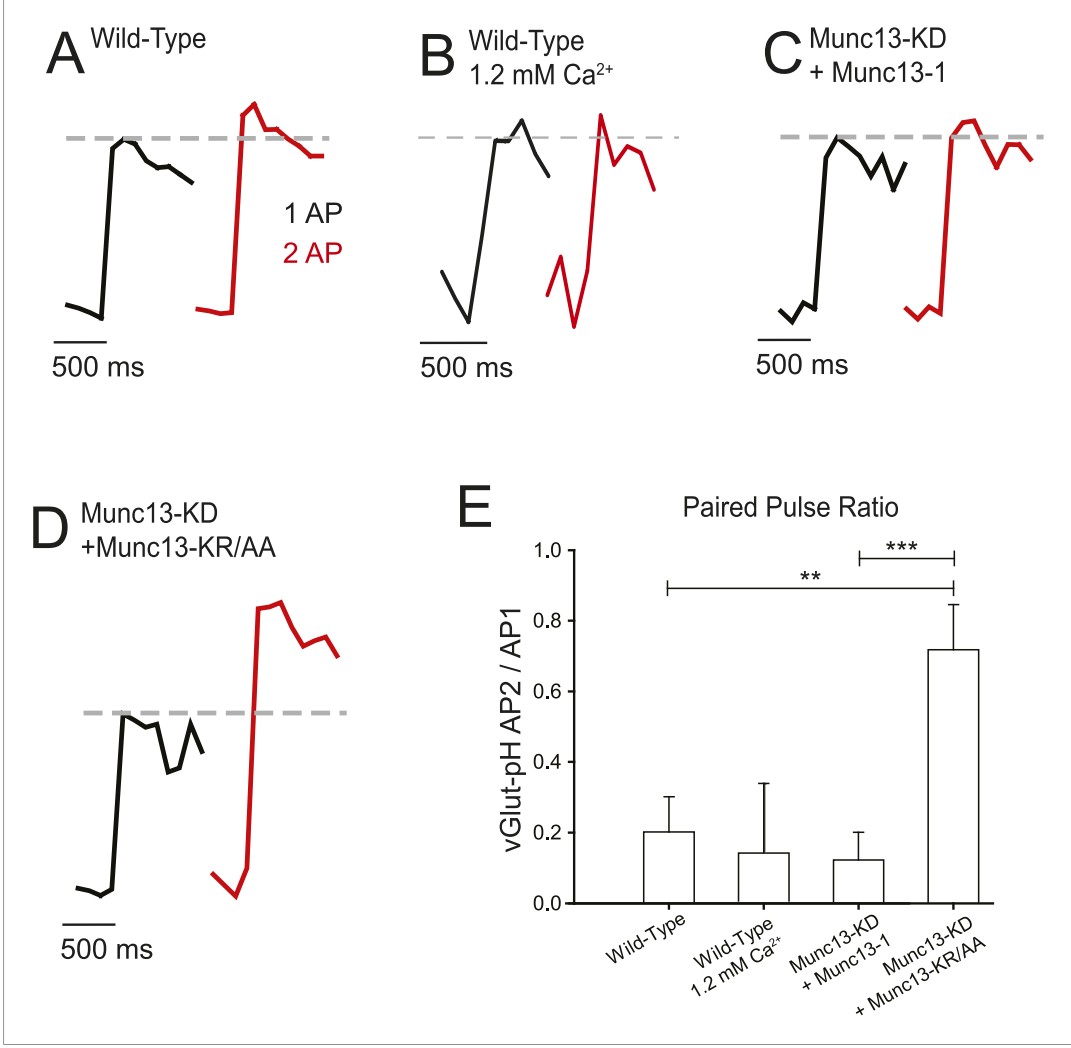

**Figure 7**. Munc13-KR/AA rescues synaptic vesicle exocytosis and reveals the effect of fast reactivation on exocytosis. (**A**–**C**) Average traces synaptic vesicle exocytosis detected by vGlut1-pHluorin in response to single APs (black) and two APs separated by 2 ms (red) for wild-type (WT) in 2 mM external $Ca^{2+}$ (**A**), 1.2 mM external $Ca^{2+}$ (**B**), Munc13-KD + Munc13-1 (**C**), and Munc13-KD + Munc13-KR/AA. Dashed line shows the response amplitude of the single AP for comparison (**D**). Increased exocytosis with two APs in Munc13-KD + Munc13-KR/AA cells indicates that the trends in $Ca^{2+}$ influx have a demonstrable effect on exocytosis, and that the Munc13 interaction with VGCC affects neural processing on this time scale. Exocytosis for a single AP is depressed in Munc13-KD + Munc13-KR/AA terminal by 38% compared to WT rescue, consistent with reduced $Ca^{2+}$ influx in these cells. (**E**) Average values for fast reactivation at 2 ms ISI for the cell types in **A**–**C**. Results are mean ± SEM. **$p < 0.01$, ***$p < 0.001$, all other comparisons n.s.

there might be a 3-way partnership between these proteins that sculpts VGCC and Munc13 function. The synprint interaction motif in Munc13-3 is slightly different than that in Munc13-1 (KRRT as opposed to KKRT, *Figure 1*), potentially reflecting differences in its interaction with VGCCs and its impact on synaptic performance for pairs of AP stimuli on fast time scales. At present, a number of presynaptic proteins, including key active-zone proteins such as RIM and ELKS/bruchpilot are known to interact with VGCCs, but the interactions have been shown largely to control VGCC localization and abundance (*Kittel et al., 2006*). It will be interesting in the future to see what active-zone protein interactions are necessary to allow function of Munc13 on VGCCs. Moreover, RIM1 and synaptotagmin contain synprint interaction motifs similar to Munc13, posing the possibility that VGCC properties could be controlled by these proteins as well. The C2B domain in synaptotagmin has also been identified as

a $Ca^{2+}$-independent mode of interaction with phosphoinositide-4,5-bisphosphate ($PIP_2$) on the plasma membrane (*Bai et al., 2004*; *Kuo et al., 2009*) and potentially facilitates synaptotagmin homo-oligomerization on lipid bilayers (*Wang et al., 2014*). These observations suggest that Munc13's C2B domain could perform a similar function and that the interaction between the C2B domain and $PIP_2$ may be important in mediating the effects of Munc13 on VGCCs or may be an additional level of regulation. This possibility would contribute an additional layer of complexity to the emerging role of $PIP_2$ in the modulation of VGCC function (*Michailidis et al., 2007*; *Roberts-Crowley et al., 2009*). The recent observation that $Ca^{2+}$ channels may be highly diffusible in the presynaptic membrane (*Schneider et al., 2015*) provides another potential mechanism by which interactions with active zone proteins such as Munc13 could modulate effective $Ca^{2+}$ influx. Complex high-frequency spike bursts have been recorded from many brain regions, including hippocampus (*Kandel and Spencer, 1961*). Our data show that the temporal envelope of transmission associated with such spikes in hippocampal neurons is actively controlled by the interaction of VGCCs and Munc13 rather than determined by the limitations of the transmission machinery. The work we show here illustrates a novel type of modulation conferred by the local active-zone environment on VGCCs and synaptic function and demonstrates its importance for control of synaptic throughput.

## Materials and methods

### Plasmids and protein preparation

The Munc13-KD shRNA vector was made using the pSuper shRNA system (Oligoengine) using a RNAi-targeting sequence (CCAGAGCTTTGAGATCATC) common to both Munc13-1 (NM_022861) and Munc13-2 (NM_001042579) based on the design criteria provided in *Reynolds et al. (2004)*. Munc13-1 cDNA was kindly provided by Nils Brose, and silent mutations at the RNAi-targeting sequence (CCAATCCTTTGAGATCATC) were introduced using the Quickchange kit (Strategene, La Jolla, CA) to generate knockdown resistant Munc13-1. Munc13-KR/AA mutant was generated using the Quickchange kit to introduce the two point mutations shown in *Figure 3* into knockdown resistant Munc13-1. The $Ca_V2.2$ (NM_001082191) Synprint region (amino acids: 718–963) was cloned into the pT7.7 bacterial expression vector with the addition of a 6-His tag and T7 epitope at the N-terminus and purified by affinity to Ni-Sepharose (GE Life Sciences). Munc13-1 C2B domain (amino acids: 675–820) and Munc13-KR/AA C2B domain were expressed as GST fusions on the pGEX-KG plasmid and purified by affinity chromatography on Glutathione-Sepharose (GE Life Sciences). Tetatnus toxin light chain cDNA was kindly provided by M Dong (Harvard).

### Sample preparation and microscopy

Hippocampal CA3–CA1 regions were dissected from 1- to 3-day-old Sprague Dawley rats, dissociated, plated, and transfected as previously described (*Kim and Ryan, 2010*). All imaging experiments were performed as previously described with laser illumination and EM-CCD image collection (*Hoppa et al., 2012*) and carried out at 37°C. Images were acquired with an Andor iXon1+ (model DU-897E-CSO-#BV) camera. Solid-state diode-pumped 488 nm, 532 nm, and 640 nm lasers were shuttered using acousto-optic modulation.

### Immunofluorescence

To quantify endogenous, mutant Munc13 isoforms and Cav2.1 in presynaptic boutons, neurons expressing VAMP2-mCherry alone or with a combination of knockdown, and Munc13 mutant constructs were fixed with 4% paraformaldehyde, permeablized with 0.1% Triton X-100, and blocked for 1 hr at 37°C with 5% BSA. For Munc13 staining cells were then labeled rabbit anti-Munc13 (Synaptic Systems) at a 1:500 dilution in blocking solution overnight at 4°C, followed by treatment for 1 hr at 37°C with Alexa-488 conjugated goat anti-rabbit secondary antibody (Invitrogen) at 1:1000 dilution in blocking solution. Expression levels were determined by quantifying the Alexa-488 fluorescence over a 2-μM region of interest over presynaptic boutons in transfected neurons as identified by VAMP2-mCherry labeling. These values were corrected against local background staining and normalized to neighboring untransfected boutons to correct for variations in global staining intensity. Cav2.1 staining was carried out as described (*Hoppa et al., 2012*).

## Live cell imaging

During imaging, cells were maintained at 37°C and continuously perfused at 200 µl/min with basal buffered saline (119 mM NaCl, 2.5 mM KCl, 2 mM CaCl$_2$, 25 mM HEPES, 30 mM glucose, 10 µM, at pH 7.4) containing 10 µM 6-cyano-7-nitroquinoxaline-2,3-dione (CNQX) and 50 µM D, L-2-amino-5-phosphonovaleric acid (AP5) (all from Sigma) to suppress postsynaptic responses. Cells were depolarized with field potentials of approximately 10 V/cm for 1 ms via platinum–iridium electrodes. Exocytosis experiments were performed by repeated (once per minute) stimulation of vGpH-expressing cells with either single or double pulses at 100 Hz imaging rates and the averaging of >10 runs for each condition and resampled at 20 Hz post-acquisition. Values were then normalized to the fluorescence from exposure to NH$_4$Cl as previously described (*Hoppa et al., 2012*). Ca$^{2+}$ measurements were performed by loading cells with 1 µg fluo5F-AM (Invitrogen) at 30°C for 10 min followed by a 10 min wash and approximately 5 min equilibration at 37°C. Reactivation experiments were performed by imaging neurons at 100 Hz and stimulating with a single pulse, followed by a double pulse 1 s later as shown in *Figure 1*. All values for single APs and reactivation experiments fall within the linear response range of the probe and are normalized as the change in fluorescence to the pre-stimulation baseline fluorescence ($\Delta F/F_0$). Deviation from linearity for single AP and 2 ms ISI experiments was insignificant. Reactivation experiments were performed by stimulating a neuron with 50 APs at 100 Hz followed by stimulation with a single AP 500 ms after the end of the train. 500 Hz trains and reactivation experiments were linearized using the fluorescence following ionomycin as the value for saturated probe. Voltage imaging was performed by imaging boutons expressing Arch-GFP excited at 640 nm at a 2000 Hz imaging rate with OptoMask occlusion and isolated crop mode for a total of 390 single or double pulses at 8 stimuli/s. All 390 runs were then averaged into a single image stack, and the response for each bouton averaged together, corrected for the 500 µs relaxation time constant of ARCH in OriginLab, and reported as $\Delta F/F_0$. ARCH amplitudes for single AP were simply taken as the peak value of the $\Delta F/F_0$ in which the baseline fluorescence was corrected for background (*Hoppa et al., 2014*). We previously showed that single bouton ARCH fluorescence waveforms contain a small direct field stimulus-driven change in membrane potential which is summed linearly with the signal (*Hoppa et al., 2014*). Measurements of stimulation in the presence of TTX were thus subtracted from the signal obtained in the absence of TTX as described (*Hoppa et al., 2014*). In order to obtain a robust estimate of the ratio of the second to first AP at 30°C, it is necessary to correct for the residual signal present from the first AP at the 2 ms time point. We subtracted off the residual ARCH amplitude at the 2 ms time point from the trials in which a single stimulus was given from the ARCH amplitude at 2 ms when 2 AP stimuli were delivered. At 37°C this correction was not performed since the AP is narrower and the contamination is negligible. We estimated the FWHM of the second AP by spline fitting the falling phase of this AP and interpolating the 50% crossing point. This value was doubled to obtain the FWHM.

## Co-precipitation experiments

Glutathione-Sepharose 4B beads for co-precipitation experiments were stored in 20% ethanol. Prior to protein loading, 20 µl of beads were washed 3× in HEPES buffered saline (HBS; 50 mM HEPES, pH 7.4, 100 mM NaCl). The beads were then incubated with 30 µg of purified GST or GST-fusion protein in 200 µl HBS with 0.1% Triton X-100 for 4 hr at 4°C. Beads were washed 5× in HBS and then incubated with 30 µg purified T7-tagged synprint region in 200 µl HBS with 0.1% Triton X-100 with either 1 mM EDTA or 1 mM CaCl$_2$ overnight at 4°C. Beads were washed 5× with HBS and the bound proteins solubilized in Tris-SDS (12 mM Tris, 0.4% SDS, 1% BME, pH 6.8). After SDS-PAGE electrophoresis, protein bands were visualized by western blot with anti-T7 mAb (Novagen) and chemiluminescence.

## Statistics

GST-pulldown statistics were performed by multi-factor ANOVA against EDTA/Ca$^{2+}$, genotype, and EDTA/Ca$^{2+}$–genotype interaction. Otherwise statistics were performed by 1-tailed t-test with the Holm-Bonferroni correction for multiple comparisons.

## Acknowledgements

This work was supported by a grant from the NIH (MH085783) to TAR. We would like to thank D Eliezer and associates for assistance preparing proteins and F Maxfield and associates for assistance

with western blotting. We thank Jeremy Dittman and members of the Ryan lab for comments and Julia Marrs for technical assistance.

## Additional information

### Funding

| Funder | Grant reference | Author |
| --- | --- | --- |
| National Institutes of Health (NIH) | MH085783 | Mingyu Xue |

The funder had no role in study design, data collection and interpretation, or the decision to submit the work for publication.

### Author contributions

NC, Conception and design, Acquisition of data, Analysis and interpretation of data, Drafting or revising the article; GG, Acquisition of data, Analysis and interpretation of data, Drafting or revising the article; MX, Acquisition of data, Analysis and interpretation of data; TAR, Conception and design, Analysis and interpretation of data, Drafting or revising the article

### Ethics

Animal experimentation: This study was performed in strict accordance with the recommendations in the Guide for the Care and Use of Laboratory Animals of the National Institutes of Health. All of the animals were handled according to approved institutional animal care and use committee (IACUC) protocols (0601-450A) of Weill Cornell Medical College.

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
