## [Decision Letter]

Thank you for submitting your work entitled “The active-zone protein Munc13 controls the use-dependence of presynaptic voltage-gated calcium channels” for peer review at *eLife*. Your submission has been favorably evaluated by a Senior editor and three reviewers, one of whom is a member of our Board of Reviewing Editors. The following individuals responsible for the peer review of your submission have agreed to reveal their identity: Graeme Davis (Reviewing editor) and Josh Kaplan (peer reviewer). A further reviewer remains anonymous.

The reviewers have discussed the reviews with one another and the Reviewing editor has drafted this decision to help you prepare a revised submission. The authors provide compelling evidence that Munc13 can bind directly to Ca_V_2 (N-type) calcium channels and that this interaction alters the calcium channel function, and likely Ca_V_2 gating. For several reasons, these results are both very interesting and highly significant. Munc13 proteins have been widely studied for their role in promoting SNARE mediated fusion of synaptic vesicles (SVs). This study shows that Munc13 likely also plays a direct role in shaping the calcium transient that drives these SV fusions. Since it plays a role in both SV fusion and in shaping the calcium transient, Munc13 proteins are excellent candidates for physically coupling SV fusion to calcium entry. Finally, as the authors indicate, a large number of active zone proteins interact with Ca_V_2 channels. Thus, the Munc13 results presented here may represent a general principle by which these Ca_V_2 binding proteins alter the calcium transient and synaptic transmission. The work should be of broad interest and, in this respect, is appropriate for publication in *eLife*. However, there are a number of concerns discussed by the reviewers that should be addressed:

1) Although a recording is shown, which documents lack of spike failure at 37°C it is not clear whether the n=7 cases reported really guarantee that spike failure does not happen at the recording locations used for the 2ms data point of Figure 4. This worry is strengthened by the fact, that the large difference observed in the 2 msec data point dwindles to a statistically non-significant difference in the neighboring data point (Figure 4) with a 3 msec ISI. It may well be that the authors have other circumstantial evidence that spike failure is not a problem. These are remarkable data and, as such, deserve extra scrutiny. It is the authors’ responsibility to supply as much evidence as possible to convince the reader of this essential conclusion.

2) The KR/AA mutated Munc13-1 fails to rescue the Munc13 knockdown for single AP calcium entry, recovery of calcium entry following bursts, and inhibition of calcium entry during 2 ms ISI protocols. This is the central argument that these changes in Ca_V_2 function are caused by changes in Munc13-1 binding to the synprint site. An important control for these experiments is that the KR/AA mutant retains other Munc13 functions. To address this point, the authors show that KR/AA mutated Munc13-1 reconstitutes single AP evoked SV fusions (using vGluT/phluorin). This figure should also show that the single AP vGluT/pH signal is effectively blocked in the non-rescued Munc13 KD cells. The prior paper (29) cited for this control did not analyze single AP responses in the KD cells.

3) Is the AP wave form during 2 ms ISI stimulation altered in Munc13 KD cells? The current manuscript shows that AP wave form is not altered in WT cells during 2 ms ISI stimulation but does not control for wave form changes in the KD cells.

4) What are the height and kinetics of the second action potential measured by Arch? With this information, what is the relationship between action potential waveform and both calcium influx and presynaptic release? This information will help to understand what is happening during the second action potential when calium influx and release occurs in the Munc13 KD.

5) Quantification of calcium channel antibody staining lacks methodology and sample sizes. It would be appropriate to present the data as frequency distributions, plotting both intensity and size of the measured spots.

6) The authors state that the effects of Munc13 on calcium entry on short time scales (ms-seconds) rules out effects on Ca_V_2 trafficking. Is this a safe assumption? Is it possible that lateral diffusion in the membrane could participate? This should be discussed.

---

## [Author Response]

*1) Although a recording is shown, which documents lack of spike failure at 37°C it is not clear whether the n=7 cases reported really guarantee that spike failure does not happen at the recording locations used for the 2ms data point of*
Figure 4*. This worry is strengthened by the fact, that the large difference observed in the 2 msec data point dwindles to a statistically non-significant difference in the neighboring data point (*Figure 4*) with a 3 msec ISI. It may well be that the authors have other circumstantial evidence that spike failure is not a problem. These are remarkable data and, as such, deserve extra scrutiny. It is the authors’ responsibility to supply as much evidence as possible to convince the reader of this essential conclusion*.

We now provide the details of all 7 experiments, expressed as the ratio of the 2^nd^ to 1^st^ peak as a scatter-gram. These data demonstrate that we simply never see failure at 37°. We feel we must have confused the reviewer however with regard to the data in Figure 4. This data reports Ca entry in WT (and KD). At 2 msec there is a very robust statistically significant depression of Ca entry for the 2^nd^ AP. As we examine later times the response to the 2^nd^ AP *recovers* with ∼ 2 time scales: a fast one (∼2-3 ms) and a slow one (40-50 ms). This recovery does not represent failure as it shows we now get Ca^2+^ entry. It is likely that for the mid-point in the recovery of the first phase, i.e. the steepest part of the recovery, the differences become noisier which is what we observed.

*2) The KR/AA mutated Munc13-1 fails to rescue the Munc13 knockdown for single AP calcium entry, recovery of calcium entry following bursts, and inhibition of calcium entry during 2 ms ISI protocols. This is the central argument that these changes in Ca*_*V*_*2 function are caused by changes in Munc13-1 binding to the synprint site. An important control for these experiments is that the KR/AA mutant retains other Munc13 functions. To address this point, the authors show that KR/AA mutated Munc13-1 reconstitutes single AP evoked SV fusions (using vGluT/phluorin). This figure should also show that the single AP vGluT/pH signal is effectively blocked in the non-rescued Munc13 KD cells. The prior paper (*[29]*) cited for this control did not analyze single AP responses in the KD cells.*

We now provide vG-pHluorin recordings for single AP stimuli in Munc13 KD neurons. As with our previously published data with responses to stimulus trains, nerve terminals in these neurons show no single AP responses either.

*3) Is the AP wave form during 2 ms ISI stimulation altered in Munc13 KD cells? The current manuscript shows that AP wave form is not altered in WT cells during 2 ms ISI stimulation but does not control for wave form changes in the KD cells*.

We provide details regarding the ratios of the peak heights of the 2^nd^ versus 1^st^ AP for both WT and Munc13KD as well as estimates of the AP widths for the 2^nd^ sAP in WT and Munc13 KD. No statistically significant differences were observed.

4) What are the height and kinetics of the second action potential measured by Arch? With this information, what is the relationship between action potential waveform and both calcium influx and presynaptic release? This information will help to understand what is happening during the second action potential when calium influx and release occurs in the Munc13 KD.

See (3).

*5) Quantification of calcium channel antibody staining lacks methodology and sample sizes. It would be appropriate to present the data as frequency distributions, plotting both intensity and size of the measured spots*.

We now provide the frequency distribution for the staining for fixed ROI sizes of Munc13 KD and WT boutons. We did not explore this extensively since the antibody is against only the minor isoform of Ca_V_ channels at these synapses and is not a measure of the more relevant surface abundance. Nonetheless we provide the statistical details showing that Munc13 ablation does not alter Ca_V_2.1 staining.

*6) The authors state that the effects of Munc13 on calcium entry on short time scales (ms-seconds) rules out effects on Ca*_*V*_*2 trafficking. Is this a safe assumption? Is it possible that lateral diffusion in the membrane could participate? This should be discussed*.

We have included a discussion of the possible relevant cell biological events for a 2 msec time scale.